# Phillygenin Attenuated Colon Inflammation and Improved Intestinal Mucosal Barrier in DSS-induced Colitis Mice via TLR4/Src Mediated MAPK and NF-κB Signaling Pathways

**DOI:** 10.3390/ijms24032238

**Published:** 2023-01-23

**Authors:** Huan-Huan Xue, Jing-Jing Li, Shi-Fei Li, Jing Guo, Rui-Ping Yan, Ting-Gui Chen, Xiang-Hua Shi, Jin-Dong Wang, Li-Wei Zhang

**Affiliations:** 1Institute of Molecular Science, Modern Research Center for Traditional Chinese Medicine, Key Laboratory of Chemical Biology and Molecular Engineering of Ministry of Education, Shanxi University, Taiyuan 030006, China; 2Shanxi Institute of Medicine and Life Science, Taiyuan 030006, China

**Keywords:** Phillygenin, ulcerative colitis, transcriptomics, tyrosine kinase Src, MAPK, NF-κB, intestinal mucosal barrier

## Abstract

Ulcerative colitis (UC) is a chronic, relapsing, and nonspecific inflammatory bowel disease (IBD). Phillygenin (PHI), a natural bioactive ingredient, isolated from *Forsythiae Fructus*, exhibits anti-inflammatory, anti-oxidative, and hepatoprotective activities. However, few reports provide direct evidence on the efficacy of PHI in improving colitis mice. The present study elucidated that the symptoms of DSS-induced colitis mice were alleviated after PHI administration, including body weight loss, the disease activity index, colon length shortening, colonic pathological damage, splenomegaly, and hepatomegaly. PHI treatment improved the intestinal mucosal barrier by protecting goblet cells, promoting gene expressions of *Clca1*, *Slc26a3*, and *Aqp8*, increasing tight junction proteins (TJs), and reducing epithelial cell apoptosis. In addition, the levels of oxidative stress (MPO, SOD, and MDA) and inflammatory cytokines (TNF-α, IL-1β, IL-6, and IL-10) were reversed by PHI in colitis mice. According to transcriptome and network pharmacology analysis, inflammatory pathway might be an important mechanism for PHI to improve colitis. Western blotting displayed that the PHI inhibited the activation of tyrosine kinase Src mediated by TLR4, and then reduced the phosphorylation of downstream proteins p38, JNK, and NF-κB in colitis mice. In summary, our results suggested that PHI might be an appropriate and effective drug candidate to protect colitis.

## 1. Introduction

Ulcerative colitis (UC), a subtype of inflammatory bowel disease (IBD), is characterized by complicated and relapsing inflammation in the gastrointestinal tract [1]. Although incidence and prevalence has been continuously rising in the world, the pathogenesis of UC remains largely unclear [2]. Consequently, the treatment is far from optimal [3]. Especially UC would easily develop into colon cancer if untimely treated [4]. It was reported that the destruction of the intestinal mucosal barrier and persistent inflammation played a crucial role in the development and progression of UC [5]. Inflammatory response is the major characteristic of colon injury in IBD patients and colitis model, which is a key factor leading to intestinal barrier damage [6]. Intestinal mucosal barrier separates lumen contents from lamina propria. The damage of intestinal barrier integrity may cause the harmful adhesion and microbial invasion, which further aggravate colitis [7]. The current treatment drugs can quell the production of inflammatory signals to alleviate UC, including aminosalicylates, corticosteroids, immunomodulators, and anti-TNF-α drugs [8]. Due to the limitation of clinical efficacy and potential side effects, the long-term use of these drugs is limited [9]. For example, aminosalicylates drug mesalazine (MES) is usually used to treat mild to moderate colitis, which had detrimental side effects, such as aplastic anemia, skin rashes, and hepatotoxicity [10]. Therefore, it is urgent to develop new and safe medicine without serious side effects for treating UC.

In this sense, the traditional Chinese medicine (TCM) contain a large number of natural bioactive compounds and are emerging as promising candidates for UC prevention, such as berberine [11], ginsenoside Rb1 [12], and andrographolide [13]. *Forsythiae Fructus*, the dry fruit of *Forsythia suspensa* (Thunb.) Vahl (Oleaceae), has the effects of clearing heat, detoxifying, swelling, and dispersing knots. Phillygenin (PHI) is one of the natural bioactive ingredients found in *Forsythiae Fructus* and exhibites a variety of biological activities [14]. Especially due to its strong anti-inflammatory activity, PHI has the potential therapeutic effect on inflammation related diseases, such as pneumonia and nonalcoholic fatty liver [15,16]. Furthermore, study had shown that PHI could protect the intestinal epithelial barrier by increasing the expression of tight junction proteins (TJs) in colon tissue [17]. Therefore, we predicted that PHI may exert a beneficial effect on DSS-induced colitis mice. In this study, the reduction in inflammatory symptoms and oxidative stress, the recovery of intestinal mucosa barrier by PHI were studied in DSS-induced colitis mice. In addition, the underlying mechanisms were explored by using transcriptome and network pharmacology analysis.

## 2. Results

### 2.1. PHI Treatment Alleviated the Symptoms of DSS-Induced Colitis Mice

During the experiment, the symptoms of DSS-induced colitis mice were similar to those of human UC, such as low vitality, body weight loss, diarrhea, and occult fecal blood. The DSS-induced colitis mice exhibited progressive weight loss compared to control mice, whereas the mice that received PHI (50 and 100 mg/kg, *P* < 0.001) and MES (100 mg/kg, *P* < 0.01) showed obviously weight gain at 8 d (Figure 1B). The increased disease activity index (DAI) score and shorted colon length were also observed in colitis mice (*P* < 0.001), whereas these anomalous changes were restored significantly by PHI (25, 50, and 100 mg/kg) or MES (100 mg/kg) at 8 d (Figure 1C–E).

### 2.2. PHI Suppressed Colon Tissue Injuries of Colitis Mice

H&E staining was used to evaluate the colon histopathological injuries. As shown in Figure 2A, the mucosa of control mice was lined with intact simple columnar epithelial cells and numerous goblet cells. The crypts also appeared closely packed and occupied the whole thickness of the mucosa. By contrast, the colon of DSS-induced mice showed epithelial cells destruction, goblet cells loss, crypts distortion and even loss. Additionally, the colonic lamina propria of colitis mice was heavily infiltrated with inflammatory cells. However, these histopathological changes were significantly ameliorated with PHI (25, 50, and 100 mg/kg) or MES (100 mg/kg) treatment. The histological scores were also increased in colitis mice, while PHI (25, 50, and 100 mg/kg, *P* < 0.001) or MES (100 mg/kg, *P* < 0.01) treatment decreased the scores (Figure 2B). Among all of the groups, PHI (100 mg/kg) treatment manifested better protection against colon pathological damage.

### 2.3. PHI Treatment Relieved the Pathological Damages of Spleen and Liver of Colitis Mice

Hepatopathy and splenopathy are common extra-intestinal manifestations of IBD. In present study, the red pulp expansion, severe congestion and extensive vacuolation were observed in spleen H&E sections of DSS-induced colitis mice. These pathological injuries were significantly improved in colitis mice treated with PHI (100 mg/kg), especially the vacuolization of spleen (Figure 3A). The spleen index was also remarkably increased in colitis mice compared with control mice, while PHI treatment dose-dependently reduced the index (Figure 3B). Furthermore, liver H&E sections manifested ruptured hepatocytes, unconspicuous hepatocytes cord, and central vein expansion with inflammatory cell infiltration in colitis mice. Mice treated with PHI (100 mg/kg) were significantly associated with less liver pathological injuries (Figure 3C). The liver index also significantly increased in colitis mice compared with control mice, while PHI treatment dose-dependently decreased the index (Figure 3D). However, there were no significant improvement on splenomegaly and hepatomegaly in MES group. 

### 2.4. PHI Treatment Improved the Intestinal Mucosa Barrier of Colitis Mice

The TJs were measured to analyze the effect of PHI on intestinal epithelial barrier. The protein expressions of ZO-1, E-cadherin, and occludin were decreased in DSS-induced colitis mice, whereas PHI (25, 50, and 100 mg/kg) or MES (100 mg/kg) treatment observably increased the expressions of TJs (Figure 4A, *P* < 0.05). The expressions of ZO-1 and occludin were also up-regulated after PHI (100 mg/kg) administration by using immunohistochemistry (Figure 4B,C). Moreover, the effect of PHI on colon apoptosis in colitis mice was measured by TUNEL staining. A large number of TUNEL positive cells were observed in colon tissue of colitis mice, but PHI (100 mg/kg) treatment significantly reduced TUNEL positive cells in the colon (Figure 4D, *P* < 0.01). The number of goblet cells was measured to analyze the effect of PHI on the intestinal mucus layer. AB-PAS staining analysis revealed that the number of goblet cells per crypt in the colon tissue was significantly decreased in colitis mice compared to control mice. PHI (100 mg/kg) treatment effectively inhibited the goblet cell loss (Figure 4E, *P* < 0.001). 

### 2.5. PHI Treatment inhibited levels of Oxidative Stress and Inflammatory Cytokines

MDA and SOD in colon tissue were measured to evaluate the effect of PHI on oxidative stress in colitis mice. The higher content of MDA and lower activity of SOD were observed in colitis mice, but PHI or MES attenuated the higher oxidative stress levels (Figure 5A,B). MPO is a pro-inflammatory and pro-oxidative enzyme, which is in direct proportion to the number of neutrophils in colonic tissue of colitis [18]. The MPO activity was significantly increased in DSS-induced colitis mice compared with control mice, while PHI (25, 50, and 100 mg/kg, *P* < 0.01) or MES (100 mg/kg, *P* < 0.01) treatment significantly suppressed MPO activity (Figure 5C). In addition, the inflammatory cytokines in colon tissues were tested to evaluate the anti-inflammatory effect of PHI on colitis. The inflammatory cytokines including TNF-α, IL-1β, IL-6, and IL-10 were higher in colitis mice than those in control mice (Figure 5D–G, *P* < 0.01). PHI (25, 50, and 100 mg/kg, *P* < 0.05) or MES (100 mg/kg, *P* < 0.01) treatment significantly decreased the concentrations of inflammatory cytokines (Figure 5D–G). Consequently, PHI could reduce the levels of oxidative stress and inflammatory cytokines in colitis mice.

### 2.6. Transcriptomic Analysis Results

The above experiments showed that PHI (100 mg/kg) treatment had better protective effect on DSS-induced colitis mice than low dose PHI. Therefore, in order to further elucidate the mechanism of PHI in treatment of UC, the differential genes affected by PHI (100 mg/kg) were determined by transcriptome. According to the conditions of padj ≤ 0.05 and |FC| ≥ 2, the differentially expressed genes were selected. A total of 1081 genes were identified in control vs. DSS, of which 484 genes were up-regulated and 597 genes were down-regulated (Figure 6A). A total of 412 genes were identified in PHI vs. DSS, of which 279 genes were up-regulated and 133 genes were down-regulated (Figure 6B). A Venn diagram further screened 201 key genes that might be the PHI treatment against colitis (Figure 6C). The expressions of 201 key genes in each group was dramatically different in clustering heat map (Figure 6D). After eliminating unknown genes, the genes of *Clca1*, *Slc26a3*, and *Aqp8* with relatively high expression and significant changes were verified by qRT-PCR. The levels of these gene expressions were consistent with transcriptional profiles (Figure 6E). In addition, the biological functions of these differentially expressed genes were obtained by Gene Ontology (GO) analysis. The top 20 GO terms were screened, and most terms were related to the immune system (Figure 6F). Furthermore, in order to identify the potential pathways of PHI in the treatment of UC, the Kyoto Encyclopedia of Genes and Genomes (KEGG) pathway enrichment analysis was performed. As a result, the top 20 pathways were screened (Figure 6G). The top three pathways were “complement and coagulation cascade”, “cytokine–cytokine receptor interaction”, and “interaction of viral proteins with cytokines and cytokine receptors”, which were generally considered to be related to immune and inflammation [19,20]. The above results suggested that PHI might ameliorate the colitis mice by regulating immune and inflammatory pathways. 

### 2.7. Network Pharmacology Results

Transcriptomic analysis suggested that PHI may mainly affect the inflammatory pathway in colitis mice. To further expound the mechanism of anti-inflammatory effects of PHI in the treatment of UC, the network pharmacology was used to predict the potential targets and signaling pathways. Using bioinformatics databases, the 149 targets of PHI against UC were obtained (Figure 7A). In order to screen out the key targets, the PPI network of 149 targets was constructed. The PPI results exhibited complex interactions between targets and targets, including 149 nodes and 1780 edges (Figure 7B). The deeper color of target represents the higher degree value, indicating the more significant effect of the target in the treatment of UC by PHI. It was found that Src was one of the most relevant targets. Src is involved in the regulation of intestinal inflammation in DSS-induced colitis mice [21]. It suggested that the decrease of intestinal inflammatory response by PHI may be related to Src protein. Furthermore, the KEGG pathway analysis of 149 involved targets was further conducted (Appendix A). In addition, the top 10 signaling pathways were screened to construct a disease–component–target–pathway network (Figure 7C). The deeper purple of signaling pathway represented the higher correlation, indicating the pathway is more important in the treatment of UC by PHI. The results showed that the PI3K/AKT signaling pathway, MAPK signaling pathway, and Ras signaling pathway were the top three relevant signaling pathways (Figure 7C). Among them, MAPK is a major signaling pathway stimulated by inflammatory mediators, which is the most likely mechanism of PHI inhibiting intestinal inflammatory response. Collectively, PHI treatment reduced intestinal inflammation of colitis mice, which might be related to the MAPK signaling pathway and potential target Src.

### 2.8. PHI Regulated the Expressions of TLR4, Src, and Downstream NF-κB and MAPK Pathway Proteins in LPS-Stimulated RAW264.7 Cells

The mechanism of PHI inhibiting inflammatory response was further verified in inflammatory model of LPS-stimulated RAW264.7 cells. PHI (25–100 μg/mL) had no significant effect on the survival rate of RAW264.7 cells after exposure for 24 h (Appendix A). The levels of NO and TNF-α were examined to determine the anti-inflammatory effects of PHI in LPS-stimulated RAW264.7 cells. PHI treatment obviously decreased the release of NO and TNF-α in LPS-stimulated RAW264.7 cells (Appendix A), which was consistent with previous report [22]. The expression of iNOS was also reduced by PHI in a dose-dependent manner (Appendix A). Moreover, the expression levels of TLR4 and p-Src proteins were significantly up-regulated in LPS-stimulated RAW264.7 cells, while PHI could significantly inhibit the protein expressions (Figure 8A). In addition, MAPK and NF-κB were the essential signals in the classical inflammatory pathway. The members of MAPK family, including p38 protein, signal-regulated kinases (ERKs), and c-Jun N-terminal kinases (JNKs), were obviously phosphorylated in LPS-stimulated RAW264.7 cells, whereas PHI treatment dose-dependently reduced the phosphorylation levels (Figure 8B). The IκBα degradation and nuclear translocation of p65 are two pivotal steps in NF-κB signaling pathway. The increased phosphorylation levels of p65 and IκBα proteins in LPS-stimulated RAW264.7 cells was reduced significantly by PHI treatment (Figure 8C).

### 2.9. PHI Regulated the Expressions of TLR4, Src, and Downstream NF-κB and MAPK Pathway Proteins in DSS-Induced Colitis Mice

Similar to those results of LPS stimulated-RAW264.7 cells, PHI (100mg/kg) treatment significantly reduced the expression level of TLR4 protein and the phosphorylation levels of Src, p65, IκBα, p38, and JNK proteins in colon tissues of colitis mice (Figure 9A–C, *P* < 0.05). Nevertheless, for the ERK signaling pathway, it was found that there was no significant difference of the phosphorylation level of ERK1/2 protein between control and colitis mice. However, PHI (100mg/kg) treatment obviously increased the phosphorylation level of ERK1/2 protein (Figure 9B, *P* < 0.01). The result of expression of ERK1/2 protein in colitis mice was inconsistent with those of RAW264.7 cells.

## 3. Discussion

The pathological characteristics of UC are the presence of intestinal inflammation and intestinal barrier damage [23]. PHI had been reported to have obvious anti-inflammatory, anti-oxidative, and hepatoprotective activities [24,25]. However, the potential of PHI in the treatment of UC and its mechanisms have not been fully investigated. In consequence, the present study aimed to investigate the efficacy and underlying mechanisms of PHI in alleviating DSS-induced colitis mice. Treatment with PHI ameliorated the symptoms of colitis mice, including weight loss, colon shortening, bloody stools, and loose stools. Colon pathological injuries were also reduced by PHI, such as epithelial cells destruction, crypts destruction, and goblet cells loss. Moreover, treatment with PHI resulted in microscopic amelioration of inflammatory cells infiltration, as demonstrated by the reduction in MPO (a marker of mucosal neutrophil infiltration) activity. Therefore, our results confirmed the protective effect of PHI in the treatment of DSS-induced colitis mice, which was similar to that of MES.

Noteworthily, PHI (100 mg/kg) alleviated the splenomegaly and hepatomegaly in the DSS-induced colitis mice. Basically, splenomegaly is often observed in IBD patients, which was generally correlated with the extent of immune inflammation and anemia [26]. It meant that PHI might reduce intestinal inflammation by affecting systemic immunity, which was similar with the transcriptomic analysis. Furthermore, hepatopathy is also one of the most common extra-intestinal manifestations of IBD [27]. PHI improved DSS-induced hepatomegaly and its pathological damage, which may be due to its liver-protective activity [25]. Collectively, PHI could not only alleviate colonic injuries, but also relieve DSS-induced splenomegaly and hepatomegaly. PHI treatment might be better than MES in improving the spleen and liver damages induced by DSS. The intestinal mucosal barrier is composed of mucus layer, epithelial cells, and TJs, and its damage is a characteristic feature of UC [28]. The mucus layer is the first line of protection in the intestinal barrier, protecting the host epithelial tissue from direct contact with luminal pathogenic microorganisms [29]. Goblet cells are specialized for the synthesis and secretion of mucus, of which Muc2 and Clca1 are the main components of mucus [30]. Herein, PHI significantly increased the number of goblet cells and the gene expression of *Clca1*, thereby protecting the mucus barrier. Beneath the mucus layer, intestinal epithelial cells are tightly interconnected by TJs, forming a complete mechanical barrier [31]. TJs were mainly composed of transmembrane proteins (such as occludin and E-cadherin) and scaffold proteins (such as ZO-1) [31]. Meanwhile, the increased apoptosis of intestinal epithelial cells was observed in UC patients, and excessive apoptosis can also lead to intestinal barrier damage [32]. In this study, PHI efficaciously increased the expressions of ZO-1, E-cadherin, and occludin, and relieved the intestinal epithelial cells apoptosis, thereby enhancing the intestinal epithelial barrier function. Therefore, PHI could improve the intestinal mucosal barrier by protecting goblet cells, promoting the expressions of *Clca1* gene and TJs, and reducing epithelial cell apoptosis. The colitis-preventing effect of PHI may also be due to the protection of intestinal mucosal barrier. 

The toxicity of DSS could induce the oxidative stress and inflammation of colon tissues. Colon mucosa oxidative stress promotes the excessive release of pro-inflammatory cytokines and excessive activation of apoptosis [33]. This creates an increased pro-inflammatory cellular microenvironment that further heightens oxidative stress, leading to the destruction of the intestinal barrier integrity [33]. MDA and anti-oxidative SOD enzyme are the important indicators of lipid peroxidation and oxidative stress [34]. In the present study, PHI treatment could mitigate the increased content of MDA and reduced activity of anti-oxidative SOD enzyme in colitis mice. In addition, an imbalance between intestinal pro-inflammatory and anti-inflammatory cytokines impedes the resolution of inflammation, triggering an uncontrolled inflammatory cycle in colon [35]. Sustained increase in pro-inflammatory cytokines (such as TNF-α, IL-1β, and IL-6) causes mucosal inflammation, leading to the reduced function of intestinal epithelial TJs [36]. However, anti-inflammatory cytokines IL-10 can suppress exacerbated mucosal immune responses and maintain intestinal homeostasis [37]. Therefore, the improvement of these inflammatory cytokines may afford an important strategy for the treatment of IBD. In the present study, PHI efficaciously suppressed the expressions of TNF-α, IL-1β, and IL-6 in colitis mice. However, the content of IL-10 was significantly increased in colitis mice, which was similar with the previous study [38]. The reason of higher levels of IL-10 in colitis mice may be seen as a compensatory mechanism for colitis damage. PHI significantly restored IL-10 levels, which may be related to its suppression of intestinal inflammatory responses. Therefore, PHI manifested a beneficial effect on DSS-induced colitis mice via mitigating intestinal oxidative stress and inhibiting inflammatory responses.

The transcriptome was applied to investigate the underlying mechanisms of PHI on the intestinal functions. It was found that most functional terms of GO and KEGG enrichment analysis were generally believed to be associated with immune and inflammatory pathways (Figure 6F,G). Chemokine family genes were mainly involved in the immune and inflammatory pathways, such as *Ccl11*, *Cxcr5*, *Ccr3*, and *Ccr10*. The chemokines are related to inflammation response by contributing to immune cells migration to colonic sites, and its blockade has become a therapeutic target for colitis [39]. In addition, the genes of *Clca1*, *Slc26a3*, and *Aqp8* screened from transcriptome were verified by qRT-PCR. The secreted goblet cell protein CLCA1 (*Clca1*) is thought to act as a mucus protein to protect the intestinal mucus barrier [40]. DRA (*Slc26a3*) not only is a Cl^−^/HCO3^−^ exchanger contributed to intestinal fluid absorption and enterocyte acid/base balance, but also ensures intestinal barrier function through stabilizing TJs [41]. AQP8 (*Aqp8*), localized to the apical membrane in colonic intestinal epithelial cells, is a key regulator of intestinal fluid resorption in the intestinal mucosa, and its dysregulation drives malabsorptive diarrhea [42]. In this study, in the mice treated with PHI were observed more normal-shaped feces than that of mice treated with DSS, which may be due to the increase in intestinal fluid absorption function of DRA and AQP8 by PHI. Thus, PHI treatment could promote the gene expressions of *Clca1*, *Slc26a3*, and *Aqp8* to enhance the intestinal mucosal barrier function. However, as it is unclear how PHI regulates colitis via these three genes, further study of this is warranted in future. In short, the transcriptomic analysis suggested that PHI mainly affected the genes related to immune-inflammatory and intestinal mucosal barrier pathways to protect colitis mice.

We further explored the mechanism of PHI inhibiting inflammatory response in vitro and in vivo inflammatory models. The present study proved PHI reduced the production and release of inflammatory factors such as IL-6, IL-1β, and TNF-α in both LPS-induced RAW264.7 cells and DSS-induced colitis mice. The possible inflammatory target Src was screened through network pharmacology. It was reported that Src is a member of the highly conserved non-receptor tyrosine kinases family involved in a variety of intracellular signal transduction [43]. The studies have reported that the activation of Src kinase not only has a harmful effect on the intestinal epithelial barrier, but also plays a crucial role in the inflammatory response mechanism of colitis [44,45]. The inflammatory response triggered by LPS in macrophages is mediated by TLR4, and activation of TLR4 also triggers several protein tyrosine kinases, including Src, Syk, and SFK kinases [46]. Src is involved in LPS/TLR4 mediated inflammatory response [47]. The present results manifested that LPS stimulation increased the expression of membrane protein TLR4 and the phosphorylation of intracellular Src in RAW264.7 cells, while PHI could significantly inhibit these changes. Similarly, PHI also reversed the changes of these proteins in DSS-induced colitis mice. Our findings illustrated that PHI might inhibit the phosphorylation of tyrosine kinase Src mediated by TLR4, and then inhibiting the expression of inflammatory factors IL-6, IL-1β, and TNF-α.

Both MAPK and NF-κB are major signal messengers that regulate pro-inflammatory genes transcription in the inflammatory process. Src is one of the major protein tyrosine kinases regulating MAPK and NF-κB signals, and is an upstream component of these signaling molecules [46]. MAPK has the three major subfamilies, including p38, ERK1/2, and JNK proteins, which are mainly involved in the inflammatory signaling pathway stimulated by inflammatory mediators [48]. In addition, NF-κB is an essential transcription factor in the classical inflammatory pathway [49]. In the inactive state, NF-κB binds to the inhibitory protein IκBα in the cytoplasm. When in an active state, IκBα is phosphorylated, and the free NF-κB translocates to the nucleus and induces the production in inflammatory factors [49]. In current results, the increased phosphorylation levels of P38, JNK, ERK1/2, IκBα, and p65 were observed in LPS-stimulated RAW264.7 cells, while PHI attenuated the phosphorylation levels. Consistently, PHI decreased the phosphorylation levels of p38, JNK, IκBα, and p65 proteins in colon tissue of colitis mice. However, PHI did not inhibit the phosphorylation of ERK1/2, but enhanced its phosphorylation level. It was reported that P38 and JNK were the critical regulators of inflammation, autoimmunity, apoptosis, cancer, and neurodevelopment, and they are more related to inflammation. [48]. The ERK signaling pathway mainly regulated the cell proliferation, differentiation, meiosis, neural cell learning, and memory function according to stimulation intensity and duration [50]. Therefore, PHI might significantly block the activation of the p38, JNK, and NF-κB signaling pathway to regulate the secretion of inflammatory factors.

## 4. Materials and Methods

### 4.1. Materials and Reagents

PHI was prepared by our laboratory (purity > 98.0%). DSS (molecular weight of 36,000–50,000 Da) was obtained from MP Biochemicals (Santa Ana, CA, USA). The enzyme-linked immunosorbent assay (ELISA) kits (such as TNF-α, IL-1β, IL-6, and IL-10) and TUNEL assay kit were purchased from Boster Biological Technology (Wuhan, China). The activity assay kits of MPO, SOD, and MDA were purchased from Jiancheng Bioengineering Institute (Nanjing, China). The NO assay kit purchased from Beyotime Biotechnology (Shanghai, China). The antibodies such as p-Src (ab185617), p-p38 (ab195049), JNK (ab179461), p-JNK (ab124956), ERK1/2 (ab184699), p-ERK1/2 (ab201015), p65 (ab32536), p-p65 (ab76302), IκBα (ab76429), p-IκBα (ab133462), occludin (ab216327), E-cadherin (ab231303), and ZO-1 (ab190085) were purchased from Abcam (Cambridge, UK). The other antibodies of iNOS (A3200), TLR4 (A5258), and Src (A0324) were purchased from ABclonal Technology (Wuhan, China). The antibody p38 (D155224) and the MightyScript Plus First Strand cDNA Synthesis Master Mix were obtained from Sangon Biotechnology (Wuhan, China). SYBR green mix was purchased from Mei5 Biotechnology (Beijing, China). Triol reagent was obtained from Takara (Shiga, Japan).

### 4.2. Experimental Animals 

Seven-week-old male BALB/c mice (weight 18-22 g) were obtained from the Vital River Laboratory Animal Technology Co., Ltd. (Beijing, China). The animals were housed in animal cages with room temperature and a 12 h light/dark cycle throughout the experimental period. All the experiments were conducted under the guidelines of ethic regulations of Shanxi University (Approval No. SXULL2022018).

### 4.3. DSS-Induced Colitis Mice and PHI Administration

Experimental colitis was induced by giving mice drinking water containing 3% (*w*/*v*) DSS for 8 d. Figure 1A showed the experimental procedure. Briefly, all the mice were received food and water ad libitum for 7 d to adapt themselves to the environment before the experiments. Then, the mice were randomly divided into 6 groups (*n* = 10 per group), including control group, DSS group, MES group (100 mg/kg), DSS + PHI groups (25, 50, or 100 mg/kg). Only the control group received normal drinking water without DSS. The DSS water was changed every two days.

### 4.4. Assessment of Colitis

During DSS treatment, the changes of weight loss, stool consistency, and occult blood were recorded daily to calculate the DAI. The DAI scoring criteria is as follows: body weight loss: 0 for within 1% weight loss; 1 for 1–5% weight loss; 2 for 5–10% weight loss; 3 for 10–15% weight loss; 4 for weight loss >15%. Stool consistency: 0 for normal stool; 2 for soft stool; 4 for watery diarrhea. Fecal occult blood: 0 for normal stool; 2 for moderate; 4 for gross bleeding [51]. The average of three scores was expressed as the DAI. 

### 4.5. Samples Collection

By the end of experiment, mice were sacrificed by cervical dislocation. The whole colons were collected, and the lengths were measured and photographed. The spleens and livers of mice were taken, and their wet weights were measured. The spleen index was calculated as wet weight of spleen (mg) × 10/body weight (g) × 100%. The liver index was calculated in the same way.

### 4.6. Histopathological Analysis

The tissues of colon, spleen, and liver were further excised to evaluate microscopic damage. All tissues were immediately fixed with 10% formaldehyde and embedded in paraffin. Sections (5-μm-thick) were stained with hematoxylin and eosin (H&E), and then the sections were examined under a light microscopy for evaluating the histopathologic changes. Histological injury of colon tissues was assessed by a combined score according to the previous method [52]. 

### 4.7. Immunohistochemistry

For immunohistochemical staining, the sections (5-μm-thick) of colonic tissues were deparaffinised and rehydrated, and incubated with inhibitor of the endogenous peroxidase. Then, the sections were processed by microwave antigen retrieval and blocked for 30 min using 5% BSA blocking solution. After blocking, the sections were incubated overnight with primary antibodies against occludin (dilution 1:200) and ZO-1 (dilution 1:300) at 4 °C. Further, the sections were incubated with polyperoxidase-anti-rabbit IgG antibody at room temperature for 30 min. Chromogen diaminobenzidine (DAB) was used to stain the tissues. Finaly, all samples were observed and photographed with BX51 light microscopy (Olympus, Tokyo, Japan).

### 4.8. TUNEL Staining

TUNEL staining was performed with the apoptosis kit to detect apoptotic cells in colon sections according to the instructions of manufacturer. Those with brown-yellow nuclei were considered positive apoptotic cells. TUNEL positive cells were counted from images of at least 3 different areas of each section using Image-Pro Plus software 6.0. The percentages of TUNEL positive cells relative to total cells per sample were calculated. 

### 4.9. Periodic Acid Schiff and Alcian Blue Staining

Fresh colon was fixed in 10% buffered formalin, embedded in paraffin, sectioned, and stained with periodic acid schiff and alcian blue (AB-PAS). Under the AB-PAS staining, the goblet cells were dark blue, and the rest of the tissues were nearly colorless. The number of goblet cells was normalized to the number of crypt units. 

### 4.10. Determination of Cytokines

The fresh colon tissues were placed in normal saline and homogenized with homogenizer. Then, the oxidative stress levels (MPO, SOD, and MDA) and the inflammatory cytokines (TNF-α, IL-1β, IL-6, and IL-10) were determined according to the assay kit methods.

### 4.11. Transcriptomic Analysis

Total RNA of colon tissue was isolated and purified by using trizol reagent. The concentration and purity of the total RNA were detected by using Nanodrop2000 (Thermo, Waltham, CM, USA). A total amount of 1.0 μg RNA per sample was used for the construction of RNA-Seq libraries. Libraries were constructed using the TruSeq™ RNA Sample Prep kit (Illumina, San Diego, CA, USA) according to the instructions of manufacturer. Sequencing of the library preparations was performed on an Illumina NovaSeq 6000 sequencer by Majorbio Biopharm Biotechnology (Shanghai, China). The data were analyzed on the free online Majorbio Scientific Research Service Platform (http://www.majorbio.com (accessed on 15 June 2022)). 

### 4.12. Real-Time Quantitative Reverse Transcription Polymerase Chain Reaction 

The RNA extracted from colon was reverse-transcribed into complementary cDNA according to the instructions of manufacturer. The Real-Time Quantitative Reverse Transcription Polymerase Chain Reaction (qRT-PCR) analysis was performed using the CFX96 PCR System (BIO-RAD, Hercules, CA, USA) with the SYBR green mix. The mouse primer sequences of *Scl26a3*, *Aqp8*, *Clca1*, and *β-Actin* (reference gene) are listed as follows: *Slc26a3*, forward, 5′-TCA ACA TCC CTC CAG CCT ACG G-3′ and reverse, 5′-CGA CAA CTC CCA CCA TCA TAC TCA G-3′; *Aqp8*, forward, 5′-GGA ACA TCA GCG GTG GAC ACT TC-3′, and reverse, 5′-GGG AAT TAG CAG CAT GGT CTT GAG G-3′; *Clca1*, forward, 5′-TGC CGA TGC CAC CAA GAA TGA TG-3′ and reverse, 5′-GTG CTG CTC TCT GTC TGT CTG AAG-3′; *β-Actin*, forward, 5′-CCC GCA AAT GCT CTA AAC C-3′ and reverse, 5′-CCA ATC CTG TCT TGT TTT ATG C-3′.

### 4.13. Network Pharmacology Analysis

Target genes of PHI were screened online using databases of PharmMapper (http://www.lilab-ecust.cn/pharmmapper/ (accessed on 25 October 2022)), ChemMapper (http://lilab-ecust.cn/chemmapper/ (accessed on 25 October 2022)), swisstarget (http://www.swisstargetprediction.ch/ (accessed on 25 October 2022)) and TCMSP (http://lsp.nwu.edu.cn/tcmsp.php (accessed on 25 October 2022)). The SMILES information of PHI was obtained from the PubChem (https://pubchem.ncbi.nlm.nih.gov/ (accessed on 25 October 2022)) database. After removing duplicates, the UNIPROT database was used (https://ebi16.uniprot.org/ (accessed on 25 October 2022)) to convert the query results into the abbreviations of the corresponding human genes. In order to find targets related to UC in the category of protein coding, the GeneCards (http://www.genecards.org/ (accessed on 18 October 2022)), Therapeutic Targets Database (TTD, https://db.idrblab.org/ttd/ (accessed on 18 October 2022)), Online Mendelian Inheritance in Man (OMIM, http://www.ncbi.nlm.nih.gov/omim (accessed on 18 October 2022)) and PharmGkb (https://www.pharmgkb.org/ (accessed on 18 October 2022)) databases were searched with “Ulcerative Colitis” as a keyword. Protein–protein interaction (PPI) network was constructed by the STRING (https://string-db.org/ (accessed on 30 October 2022), version. 11.0) database. KEGG pathway enrichment analysis of target genes was conducted by DAVID database (https://david.ncifcrf.gov/home.jsp (accessed on 30 October 2022)).

### 4.14. RAW264.7 Cell Culture and PHI Treatment

The RAW264.7 cell line was originated from American Type Culture Collection (ATCC, Rockefeller, MD, USA). RAW264.7 cells were cultured in Dulbecco’s modified Eagle’s medium containing 10% heat inactivated fetal bovine serum (FBS), 100 U/mL streptomycin, and 100 U/mL penicillin at the condition of 37°C and 5% CO_2_. RAW264.7 cells were treated with PHI (25.0, 50.0, and 100.0 μg/mL) for 1 h, and then exposed to LPS (1.0 μg/mL) 6 h or 24 h. Then the levels of NO and TNF-α were determined according to the kit methods after LPS- stimulated for 24 h. In addition, the expressions of inflammatory proteins were detected by Western blotting after LPS- stimulated for 6 h.

### 4.15. Cell Viability Assay

RAW264.7 cells were seeded at a density of 2 × 10^5^ cells/mL on a 96-well cell culture plate. In addition, the cells were treated with 100 μL DMEM solutions containing different concentrations of PHI (25.0, 50.0, and 100.0 μg/mL) for 24 h. The cytotoxic effects of PHI were determined using an MTT assay, as previously described [22].

### 4.16. Western Blotting Assay

The fresh colons and RAW264.7 cells were lysed by RIPA lysate to obtain the whole protein. Then, the proteins were quantified by BCA kit and collected after heating in boiling water for 5 min. Equal amounts of protein samples were loaded to electrophoresis on an 8%–10% SDS-PAGE gel followed by transferring onto PVDF membranes. After blocking with skim milk for 90 min at room temperature, the PVDF membranes were then shaken overnight with primary antibodies at 4 °C. PVDF membranes were incubated with horseradish peroxidase (HRP)-conjugated goat anti-rabbit (or anti-mouse) IgG antibody (dilution 1:10,000) at room temperature for 1.5 h. After washing with TBST, the immunoreactive protein bands were visualized using enhanced chemiluminescence assay kit. Optical density from Western blotting assay was quantified with Image-Pro Plus software 6.0. 

### 4.17. Statistical Analysis

Statistical analyses of the data were performed using SPSS 16.0. Differences between the mean values of normally distributed data were assessed with a one-way analysis of variance (ANOVA) and the two-tailed Student’s *t* test. The value of *P* < 0.05 was considered statistically significant. All data were presented as mean ± standard error of the mean (SEM).

## 5. Conclusions

In summary, PHI effectively alleviated DSS-induced colitis mice through improving intestinal mucous layer, TJs, and epithelial cell apoptosis, reducing oxidative stress and intestinal inflammation. In terms of mechanism, PHI inhibited the phosphorylation of tyrosine kinase Src mediated by TLR4, and then reduced the activation of its downstream pathway involving p38, JNK, and NF-κB, thereby reducing intestinal inflammation of colitis mice. Notably, PHI and MES have similar protective effects on colitis, while PHI has greater improvement effect on liver and spleen damage caused by DSS. It meant that PHI may have the potential to be a more appropriate and safer candidate drug for treating UC.

## Figures and Tables

**Figure 1 ijms-24-02238-f001:**
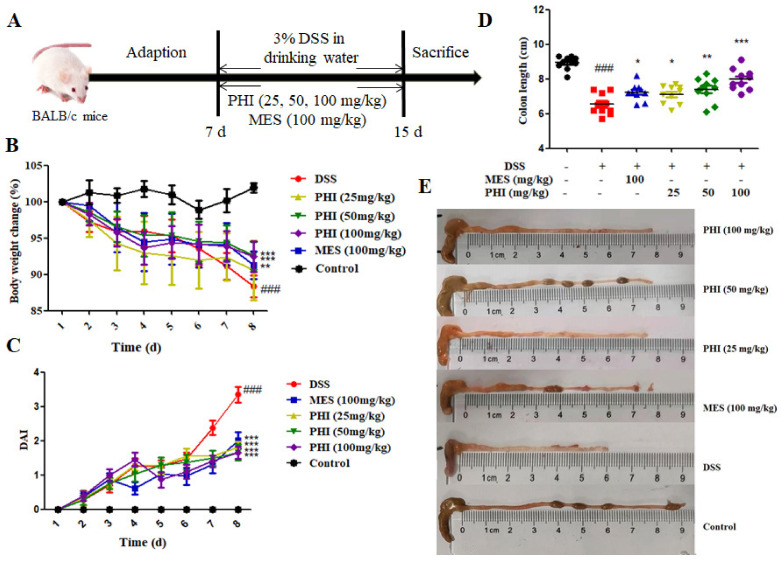
PHI alleviated the symptoms in DSS-induced colitis mice. (**A**) Experimental design. (**B**) Bodyweight changes from 1–8 d. (**C**) DAI scores from 1–8 d. (**D**) Colon lengths at 8 d. (**E**) Macroscopic pictures of colons. ^###^
*P* < 0.001 vs. control group. * *P* < 0.05, ** *P* < 0.01, and *** *P* < 0.001 vs. DSS group. (*n* = 10 mice per group).

**Figure 2 ijms-24-02238-f002:**
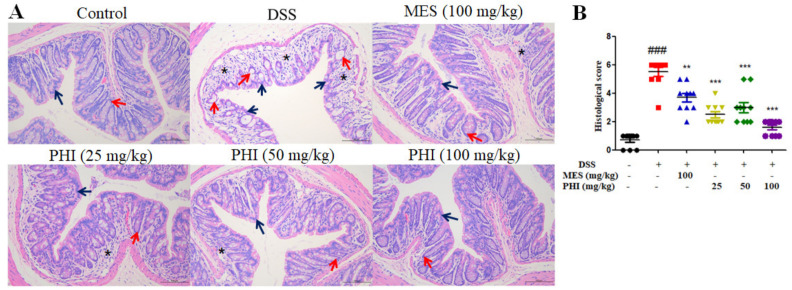
PHI improved the pathological damages of colon tissues. (**A**) Representative colon histological sections (scale bar = 100 μm). (**B**) Colonic histological score. Asterisks referred to inflammatory cell infiltration. Blue arrows pointed to columnar epithelial cells while red arrows referred to crypts. ^###^
*P* < 0.001 vs. control group. ** *P* < 0.01, and *** *P* < 0.001 vs. DSS group. (*n* = 10 mice per group).

**Figure 3 ijms-24-02238-f003:**
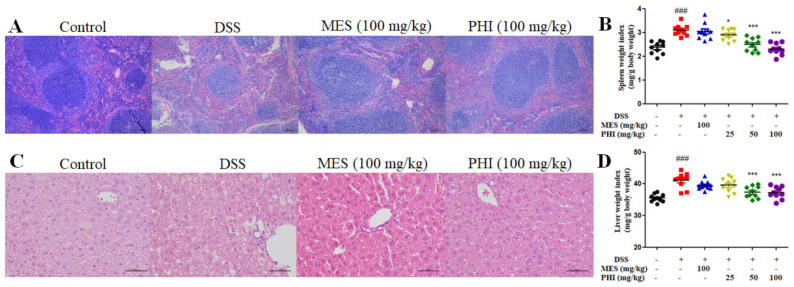
PHI improved the pathological damages of spleen and liver. (**A**) Representative spleen histological sections (scale bar = 100 μm). (**B**) Spleen index of each group mice. (**C**) Representative liver histological sections (scale bar = 50 μm). (**D**) Liver index of each group mice. ^###^
*P* < 0.001 vs. control group. * *P* < 0.05and *** *P* < 0.001 vs. DSS group. (*n* = 10 mice per group).

**Figure 4 ijms-24-02238-f004:**
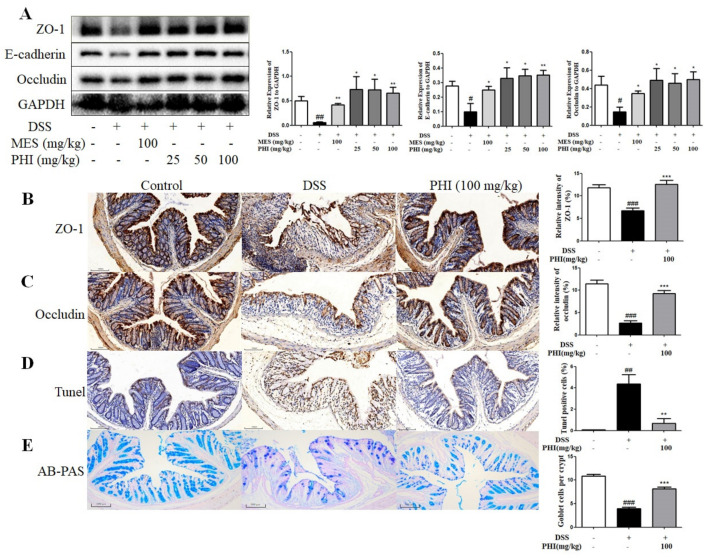
PHI treatment improved the intestinal mucosa barrier of colitis colon tissues. (**A**) Protein levels of ZO-1, E-cadherin, and occludin in colons were detected by Western blotting. (**B**,**C**) Immunohistochemical staining of ZO-1 and occludin (scale bar = 100 μm). (**D**) Apoptosis of colon tissues were subjected to TUNEL staining (scale bar = 100 μm). (**E**) Goblet cells were detected by AB-PAS staining (scale bar = 100 μm). ^#^
*P* < 0.05, ^##^
*P* < 0.01, and ^###^
*P* < 0.001 vs. control group. * *P* < 0.05, ** *P* < 0.01, and *** *P* < 0.001 vs. DSS group. (*n* = 3–6 mice per group).

**Figure 5 ijms-24-02238-f005:**
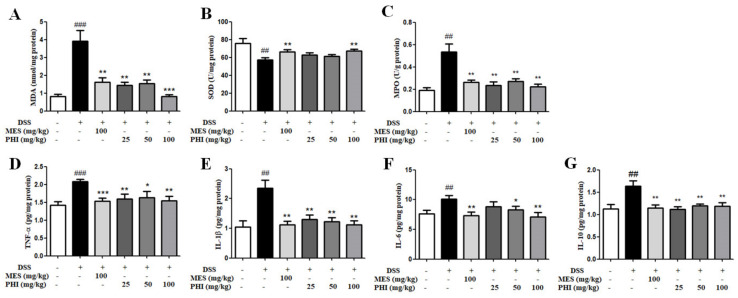
PHI inhibited the levels of oxidative stress and inflammatory cytokines. The oxidative stress levels (**A**) MDA, (**B**) SOD, and (**C**) MPO in colon tissues were determined according to the assay kit methods. The inflammatory cytokines (**D**) TNF-α, (**E**) IL-1β, (**F**) IL-6, and (**G**) IL-10 in colon tissues were determined by ELISA. ^##^
*P* < 0.01, and ^###^
*P* < 0.00 1 vs. control group. * *P* < 0.05, ** *P* < 0.01, and *** *P* < 0.001 vs. DSS group. (*n* = 10 mice per group).

**Figure 6 ijms-24-02238-f006:**
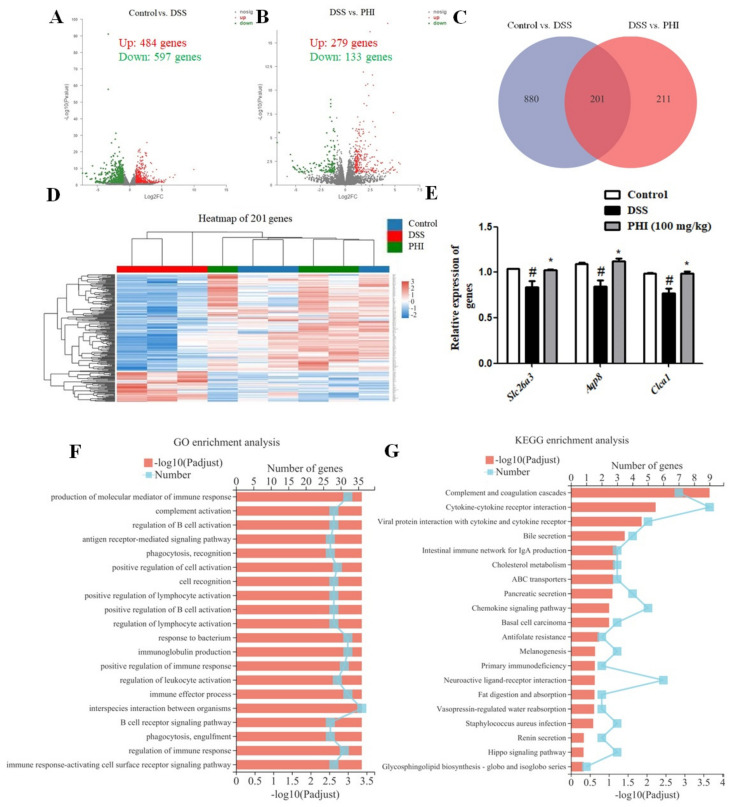
PHI (100 mg/kg) treatment altered transcriptional profiles of colitis mice. (**A**) Genes volcano map of control vs. DSS. (**B**) Genes volcano map of DSS vs. PHI. (**C**) Venn diagram of genes. (**D**) The clustering heat maps of 201 genes. (**E**) Genes were validated by using qRT-PCR. (**F**) Results of GO enrichment analysis. (**G**) Results of KEGG enrichment analysis. ^#^
*P* < 0.05 vs. control group. * *P* < 0.05vs. DSS-induced group (*n* = 3 mice per group).

**Figure 7 ijms-24-02238-f007:**
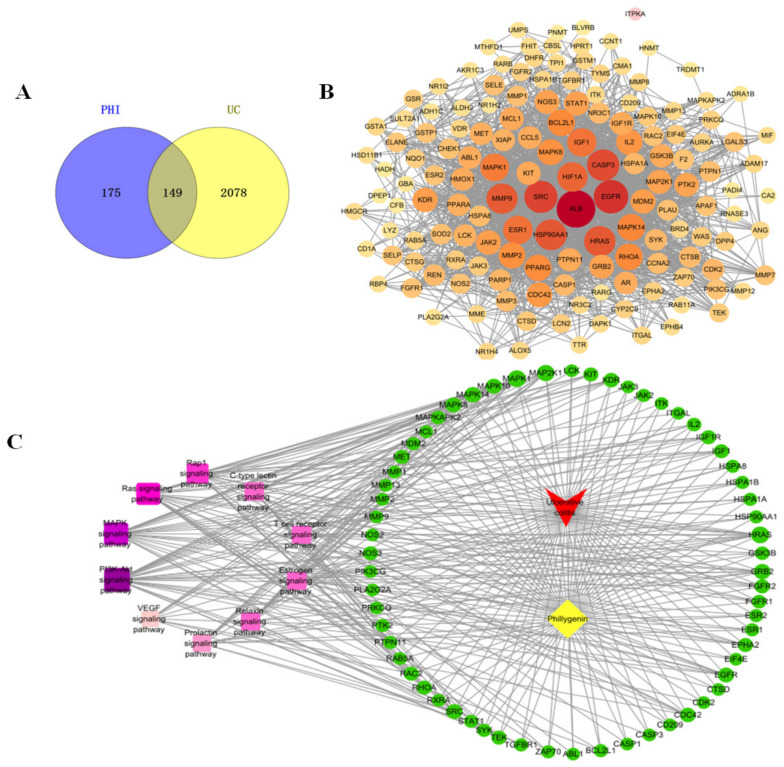
Network pharmacology analysis. (**A**) Venn diagram of potential targets of PHI against UC. (**B**) PPI network of 149 potential targets. (**C**) Disease-component-target-pathway network. PHI was marked in yellow, UC in red, targets in green, and signaling pathways in purple.

**Figure 8 ijms-24-02238-f008:**
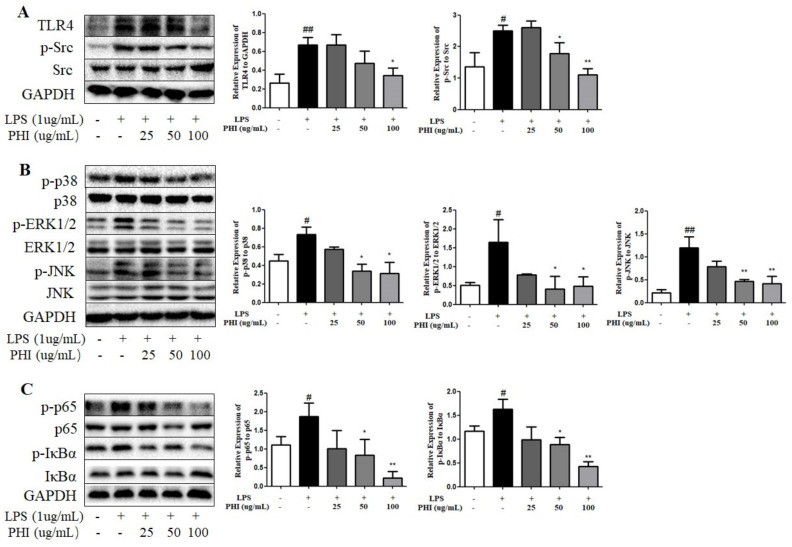
PHI regulated the expressions of TLR4, Src, and downstream NF-κB and MAPK pathway proteins in LPS-stimulated RAW264.7 cells. (**A**) Protein levels of TLR4, Src, p-Src, (**B**) p38, p-p38, JNK, p-JNK, ERK1/2, p-ERK1/2, (**C**) p65, p-p65, IκBα, and p-IκBα in RAW264.7 cells were detected by Western blotting. ^#^
*P* < 0.05and ^##^
*P* < 0.01 vs. control group. * *P* < 0.05and ** *P* < 0.01 vs. LPS-stimulated group. (*n* = 3 per group).

**Figure 9 ijms-24-02238-f009:**
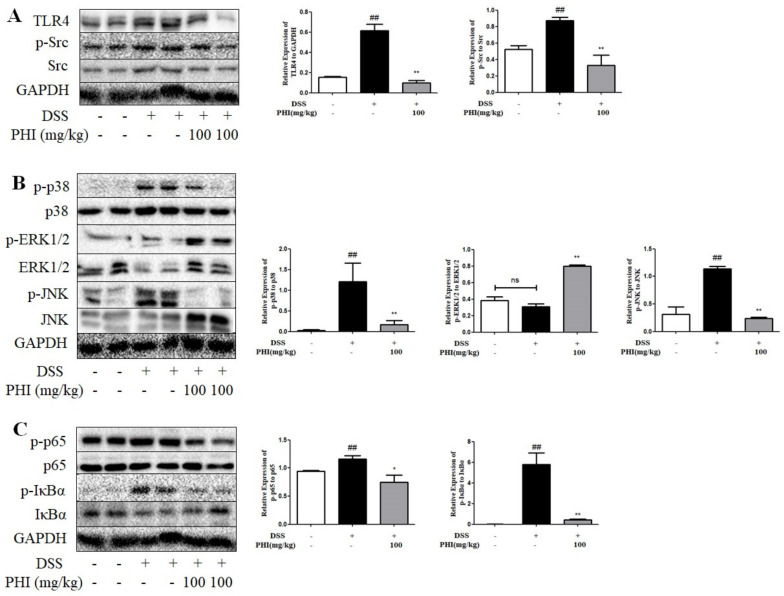
PHI treatment affected the expressions of TLR4, Src, and downstream NF-κB and MAPK pathway proteins in DSS-induced colitis mice. (**A**) Protein levels of TLR4, Src, p-Src, (**B**) p38, p-p38, JNK, p-JNK, ERK1/2, p-ERK1/2, (**C**) p65, p-p65, IκBα, and p-IκBα in DSS-induced colitis mice were detected by Western blotting. ^##^
*P* < 0.01 vs. control group. * *P* < 0.05and ** *P* < 0.01 vs. DSS group. “ns” represented not significant. (*n* = 4 mice per group).

## Data Availability

The raw data supporting the conclusions of this manuscript will be made available by the authors, without reservation, to any qualified researcher.

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
