# Peer review of "Phillygenin Attenuated Colon Inflammation and Improved Intestinal Mucosal Barrier in DSS-induced Colitis Mice via TLR4/Src Mediated MAPK and NF-κB Signaling Pathways"

_ijms, 2023, doi:10.3390/ijms24032238_

Round 1

Reviewer 1 Report

Overall, I found the topic of this manuscript interesting and innovative, proposing phillygenin (PHI) as an appropriate drug candidate to protect individuals with colitis. In addition, the results presented in this article are consistent and concrete, and the study design is well-constructed, providing high scientific soundness.

I only have several minor concerns/comments listed below:

1. This study used a mesalamine (MES) treatment group as the positive control. I assume it is because that MES is a medication commonly used to treat inflammatory bowel disease (IBD). Since the authors suggest that PHI could be a drug candidate for treating IBD, I expect to see some discussion comparing MES and PHI treatment and why the latter is preferred in some cases. Also, a short introduction about MES treatment and its disadvantages could be included at the beginning of the article, hence the need to find an alternative medicine.

2. Figure 2A - the authors described their results between lines 87-89 that "the mucosa was lined with intact simple columnar epithelial cells... the crypts also appeared closely packed...". I suggest highlighting or pointing these histological features with arrows or arrowheads in figure 2a. It's trivial, however, essential for readers unfamiliar with colon histology.

3. Figure 2A - similarly, for lymphocyte infiltration and polymorphonuclear leucocytes, as described between lines 91-93, I suggest highlighting them with arrows or arrowheads in figure 2a. 

4. Figure 4D - cell apoptosis was measured via TUNEL staining. According to the methodology (line 474), the raw number of TUNEL-positive cells was counted. However, the figures represent the final data in % format. Please consider adding a description of how the percentage was calculated.

5. Figure 4E -  data presented in this figure is the raw number of cells. It needs to be normalized to the total epithelial cells captured/considered or to an internal control such as crypt numbers. 

6. Results 2.6 - please specify the PHI concentration used in the comparison. I assume it is 100mg/kg; however, it is not mentioned anywhere in this section. Considering that three concentration groups were used in the article, it is essential to elucidate which one was used for this specific analysis and why.

Besides these, I do not have any major concerns.

Reviewer 2 Report

This study investigates the effect of phillygenin on UC. Phillygenin is an bioactive ingredient from F. Fructus, which exhibits anti inflammatory, anti-oxidative, and hepatoprotective properties. The results reported in the present study indicate that phillygenin administration in the diet alleviates the symptoms of dextran sulfate sodium (DSS)-induced colitis in male BALB/c mice, including body weight loss, colon length shortening, colon damage, splenomegaly and hepatomegaly.  Following the treatment with phillygenin the intestinal mucosal barrier integrity improved, the tight junction integrity improved, and epithelial cell apoptosis decreases  Consistent with this histological perspective, the levels of oxidative stress and inflammation (assessed as inflammatory cytokines) decreased. Western Blot analysis indicates that PHI inhibited the activation of tyrosine kinase Src mediated by TLR-4, and reduced the phospohorylation of p38, and JNK, and the translocation of NF-kB to the nucleus. 
